# Clinical Application of Cardiac Magnetic Resonance in ART-Treated AIDS Males with Short Disease Duration

**DOI:** 10.3390/diagnostics12102417

**Published:** 2022-10-06

**Authors:** Keke Hou, Hang Fu, Wei Xiong, Yueqin Gao, Liqiu Xie, Jianglin He, Xianbiao Feng, Tao Zeng, Lin Cai, Lei Xiong, Nan Jiang, Min Jiang, Bin Kang, Haiyan Zheng, Na Zhang, Yingkun Guo

**Affiliations:** 1Department of Radiology, Public Health Clinical Center of Chengdu, Chengdu 610061, China; 2Key Laboratory of Obstetric & Gynecologic and Pediatric Diseases and Birth Defects of Ministry of Education, Department of Radiology, West China Second University Hospital, Sichuan University, 20# South Renmin Road, Chengdu 610017, China; 3Department of Ultrasound, Public Health Clinical Center of Chengdu, Chengdu 610061, China; 4Department of Infectious Disease, Public Health Clinical Center of Chengdu, Chengdu 610061, China

**Keywords:** acquired immune deficiency syndrome, CMR, cardiovascular complications

## Abstract

Cardiac complications are common in antiretroviral therapy-treated (ART-treated) acquired immune deficiency syndrome (AIDS) patients, and the incidence increases with age. Myocardial injury in ART-treated AIDS patients with a relatively longer disease duration has been evaluated. However, there is no relevant study on whether patients with a short AIDS duration have cardiac dysfunction. Thirty-seven ART-treated males with AIDS and eighteen healthy controls (HCs) were prospectively included for CMR scanning. Clinical data and laboratory examination results were collected. The ART-treated males with AIDS did not have significantly reduced biventricular ejection fraction, myocardial edema, or late gadolinium enhancement. Compared with the HCs, the biventricular volume parameters and left ventricle myocardial strain indices in ART-treated males with AIDS were not significantly reduced (all *p* > 0.05). ART-treated males with AIDS were divided into subgroups according to their CD4+ T-cell counts (<350 cells/μL and ≥350 cells/μL) and duration of disease (1–12 months, 13–24 months, and 25–36 months). There was no significant decrease in left or right ventricular volume parameters or myocardial strain indices among the subgroups (all *p* > 0.05). In Pearson correlation analysis, CD4+ T-cell counts were not significantly correlated with biventricular volume parameters or left ventricular myocardial strain indices. In conclusion, ART-treated males with AIDS receiving ART therapy with a short disease duration (less than 3 years) might not develop obvious cardiac dysfunction as evaluated by routine CMR, so it is reasonable to appropriately extend the interval between cardiovascular follow-ups to more than 3 years.

## 1. Introduction

Acquired immune deficiency syndrome (AIDS) is a serious global public health concern. In AIDS patients, mortality rates have decreased and survival duration has increased with increased use of antiretroviral treatment (ART). Therefore, the management of HIV/AIDS has become a long-term process. During this long process, multiple factors, including myocardial viral infections, immune activation, inflammation, metabolic abnormalities, nutritional deficiency, and so on, have been proposed to have negative impacts on the heart and to cause cardiac complications [1]. Except for a strong association between HIV and atherosclerosis, myocardial diseases, especially dilated cardiomyopathy and myocarditis, have been reported frequently [2]. Previous studies reported that the incidence of cardiac complications in AIDS patients is approximately 25–75%, and the incidence increases as AIDS patients age [3,4,5,6]. In addition, a review concluded that AIDS patients affected by cardiac complications always had a poor prognosis [7]. AIDS-related symptomatic heart failure will become one of the leading causes of HIV/AIDS deaths worldwide in the future [8]. Therefore, timely evaluation of cardiac complications is of paramount significance in AIDS patients.

Echocardiography is the main modality for evaluating cardiac complications in AIDS patients in long-term follow-up with a relatively lower cost. However, due to the intrinsic limitations of echocardiography, such as a lower accuracy of cardiac function evaluation, poor repeatability, and an inability to characterize the myocardium, cardiac magnetic resonance (CMR) is often recommended for the assessment of myocardial involvement and accurate function evaluation in AIDS patients, especially in those in whom cardiac complications are suspected and those having clinical symptoms. A CMR study indicated that HIV infection treated with ART is more likely to result in changes in myocardial structure and function and a higher prevalence of subclinical myocardial edema, myocardial fibrosis, and frequent pericardial effusions [9]. Holloway and colleagues found that cardiac steatosis and fibrosis are highly associated with cardiac dysfunction, cardiovascular morbidity, and mortality in HIV patients [10]. Early evaluation of cardiac dysfunction could provide objective evidence for timely intervention. However, the patients included in the above studies have different and heterogeneous disease courses, with relatively longer disease courses in general. There is no unified conclusion on the time point of cardiac complication onset in patients receiving ART treatment. In addition, the number of AIDS cases diagnosed in males each year is significantly higher than that in females, and males have a more favorable response to ART treatment and have a higher cardiovascular risk than females [11,12,13,14]. Therefore, in this study, we aimed to perform an early evaluation of cardiac complications by CMR in male AIDS patients treated with ART with a short disease duration.

## 2. Materials and Methods

### 2.1. Study Population

After approval by the Ethics Committee of Chengdu Public Health Clinical Medical Center, 39 male human immunodeficiency virus (HIV)-positive patients diagnosed with AIDS were prospectively enrolled and underwent enhanced CMR examination. Inclusion criteria were as follows: (1) 18 years < age < 70 years; (2) disease duration ≤ 36 months; (3) ART therapy initiation after diagnosis; (4) sinus heart rate and no cardiovascular history, including congenital heart disease, heart valve disease, cardiomyopathies, coronary heart disease, and so on; and (5) no malignant tumors or non-HIV-related respiratory diseases. Exclusion criteria were as follows: (1) CMR contraindications; (2) acute kidney injury or severe chronic kidney disease (GFR < 30 mL/min/1.73 m^2^); and (3) poor image quality impeding postprocessing. Finally, 2 patients who could not hold their breath, which resulted in poor image quality, were excluded, and 37 patients were included. Eighteen age-matched healthy males were enrolled as healthy controls (HCs). All participants signed informed consent forms. Basic information and laboratory data were collected.

### 2.2. CMR Scanning Protocol and Imaging Postprocessing

All participants underwent a CMR scan using a 1.5-T whole-body scanner (Signa HDxt; GE Medical Systems, United States of America) in the supine position. Image acquisition was performed during the breath-holding period at the end of inspiration. A series of 8–12 continuous short-axis views of the left ventricle (LV) from the mitral valve to the level of the LV apex were obtained using steady-state free-precession sequences with the following parameters: repetition time, 4 ms; echo time, 2 ms; slice thickness, 6.0–8.0 mm; flip angle, 39°; field of view, 360 × 360 mm^2^; and matrix size, 256 × 256. Two-chamber and four-chamber long-axis cine images were also acquired. T2WI black blood images were obtained with a triple-inversion recovery sequence including a left ventricle four-chamber long-axis section and left ventricle short-axis basal, middle, and apical sections. Ten to fifteen minutes after intravenous injection of Meglumine Gadopentetate (Beilu, Beijing, China) (0.2 mL/kg body weight, flow rate of 3–5 mL/s), late gadolinium enhancement (LGE) images were obtained using the inversion recovery TrueFISP sequence (inversion time was based on TI scout) for the short-axis (slice thickness 8 mm) and 2-chamber and 4-chamber long-axis sections.

### 2.3. Data Analyses

All CMR data were analyzed using the commercially available software cvi42 (Circle Cardiovascular Imaging, Inc., Calgary, AB, Canada). The series of short-axis cine images and 2-chamber and 4-chamber short-axis cine images were loaded into the short 3D module and issue feature tracking module. Epicardial and endocardial borders were traced manually to compute cardiac function parameters and myocardial deformation parameters. Biventricular function parameters included left ventricular ejection fraction (LVEF), left ventricular end-diastolic volume (LVEDV), left ventricular end-systolic volume (LVESV), left ventricular stroke volume (LVSV), left ventricular myocardial mass (LVM), right ventricular ejection fraction (RVEF), right ventricular end-diastolic volume (RVEDV), right ventricular end-systolic volume (RVESV), and right ventricular stroke volume (RVSV). LV remodeling was defined as the ratio of LVEDV to LV mass (LVRI). Myocardial strain indices included global radial strain (GRS), global circumferential strain (GCS), global longitudinal strain (GLS), global diastolic strain rate radial (GDSR), global diastolic strain rate circumferential (GDSC), and global diastolic strain rate longitudinal (GDSL). Segmental myocardial strain indices for the basal, middle, and apical parts of the LV were also acquired.

### 2.4. Statistical Analysis

Statistical analyses were performed with SPSS (version 21.0 for Windows; SPSS, Inc., Chicago, IL, USA). Continuous variables are expressed as the mean ± standard deviation or the median and interquartile range. Categorical variables are presented as frequencies (percentages) and were compared using the chi-square test. Normal distribution was tested with the Kolmogorov–Smirnov test. Continuous variables were compared using ANOVA and independent *t* tests (normal distribution) or Kruskal–Wallis and Mann–Whitney U tests. Pearson’s test was performed to evaluate the relationship between CD4+ T-cell counts and parameters of cardiac function and myocardial strain. *p*-value < 0.05 was considered statistically significant.

## 3. Results

### 3.1. Baseline Characteristics

Thirty-seven ART-treated males with AIDS and eighteen HCs were included in this study. The baseline clinical characteristics are presented in Table 1. There was no significant difference in age, blood pressure, or body mass index between ART-treated males with AIDS and HCs (all *p* > 0.05). The mean age of ART-treated males with AIDS was 37.62 ± 11.10 years old. All patients received ART therapy, and the duration was 20.58 ± 3.72 months. Blood CD4+ T-cell counts were 358.21 ± 57.41 cells/µL. Regarding HIV-related comorbidities, we found that syphilis was present in 5 (13.51%) patients, granulocytopenia was present in 3 (8.11%) patients, HIV-related pneumonia was present in 9 (24.32%) patients, metabolic syndrome was present in 10 (27.03%) patients, and oral fungal infection was present in 2 (5.41%) patients. No patients had cardiovascular symptoms, and electrocardiography and echocardiography findings were negative in all patients. Regarding myocardial injury biomarkers, no significant abnormality was found in the myocardial enzyme spectrum or high-sensitivity c-TnT in 37 patients.

### 3.2. Comparison between the AIDS Group and the Healthy Control Group

No ART-treated males with AIDS had biventricular ejection fraction reduction, myocardial edema, or LGE. Biventricular function and LV deformation indices were compared between ART-treated males with AIDS and HCs (as shown in Table 2). Compared with the LVM of healthy controls, the LVM of ART-treated males with AIDS tended to be lower (78.14 ± 15.41 g vs. 85.26 ± 15.13 g, *p* = 0.053), while LVMi (43.37 ± 8.98 g/m^2^ vs. 45.01 ± 6.34 g/m^2^, *p* = 0.236) was similar between the groups after standardization by body surface area.

In the group of ART-treated males with AIDS, LVEF (61.75 ± 4.90% vs. 60.33 ± 4.66%, *p* = 0.307), RVEF (51.94 ± 8.75% vs. 50.89 ± 6.08%, *p* = 0.872), and LVRI (1.59 ± 0.24 mL/g vs. 1.48 ± 0.27 mL/g, *p* = 0.200) were not significantly decreased compared to those of HCs, nor were biventricular volume parameters significantly reduced (all *p* > 0.05). Figure 1 shows that the myocardial deformation parameters GRS, GCS, and GLS were not significantly different between the AIDS group and the HC group (all *p* > 0.05, see also Figure 2). Moreover, the diastolic function indices of males with AIDS, including GDSR, GDSC, and GDSL, were not significantly reduced. Segmental analysis in the basal, middle, and apical segments also did not show significant decreases.

### 3.3. Subgroup Analysis According to CD4+ T-Cell Counts

According to the CD4+ T-cell counts, ART-treated males with AIDS were divided into two groups: those with CD4+ T-cell counts < 350 cells/μL (moderate or severe reduction of immune function) and those with CD4+ T-cell counts ≥ 350 cells/μL (normal or slightly decreased immune function). The results of a comparison of cardiac function and myocardial strain are shown in Table 3. Compared with HCs, no significant decrease in biventricular ejection fraction or volume was found in ART-treated males with AIDS with different levels of CD4+ T-cell counts (all *p* > 0.05). LVRI was not significantly different in the three groups (1.54 ± 0.19 mL/g vs. 1.64 ± 0.27 mL/g vs. 1.48 ± 0.27 mL/g, *p* = 0.151). Figure 3 reports the results of the subgroup comparison: ART-treated males with AIDS with lower CD4+ T-cell counts did not have lower peak strain in any of the three directions compared to ART-treated males with AIDS with CD4+ T-cell counts ≥ 350 cells/μL and HCs (all *p* > 0.05). A diastolic function comparison showed that GDSR (-1.68 ± 0.44 1/s vs. -1.67 ± 0.84 1/s vs. -1.51 ± 0.55 1/s, *p* = 0.669), GDSC (1.02 ± 0.2 1/s vs. 1.11 ± 0.23 1/s vs. 1.01 ± 0.21 1/s, *p* = 0.294), and GDSL (0.67 ± 0.11 1/s vs. 0.7 ± 0.19 1/s vs. 0.72 ± 0.16 1/s, *p* = 0.677) were similar among the ART-treated AIDS male subgroups and HCs. Segmental analysis in basal, middle, and apical segments did not show significant decreases.

### 3.4. Subgroup Analysis According to Disease Duration

According to disease duration, the ART-treated males with AIDS were divided into three subgroups: those with a disease duration of 1–12 months, 13–24 months, and 25–36 months. There was no significant difference in the remaining left and right ventricular function and structure in these three subgroups compared to the healthy control group (shown in Table 4, all *p* > 0.05). Figure 4 shows the results of the subgroup analysis based on disease duration. None of the myocardial strain parameters were decreased in ART-treated AIDS patients with different disease durations compared to HCs (all *p* > 0.05).

### 3.5. Correlation Analysis

The results of the correlation analysis between CD4+ T-cell counts and CMR parameters are presented in Table 5. In Pearson correlation analysis, CD4+ T-cell counts were not significantly correlated with biventricular volume parameters or left ventricular myocardial strain indices (all *p* > 0.05).

## 4. Discussion

This was the first CMR study that focused on the assessment of cardiac complications in ART-treated males with AIDS with a short disease duration (within 3 years), and no obvious cardiac dysfunction was found. ART is relatively safe for the heart over a short course of treatment. We recommend that the interval between cardiac follow-ups after the diagnosis of AIDS be appropriately extended for these patients.

Previous studies that enrolled ART-treated AIDS patients with a mean disease duration of 90 months determined that AIDS patients can develop heart enlargement, ventricular septal thickening, and increased volume and myocardial mass, which further progress to diastolic and systolic dysfunction and even heart failure [15,16,17]. The results of a study on cardiac function in 156 AIDS patients with a median disease duration of 10.4 years [18] also suggested that left ventricular remodeling, revealed by an increase in LVMi (65 g/m^2^ [49–77 g/m^2^] vs. 57 g/m^2^ [49–64 g/m^2^]), is closely related to cardiovascular adverse events, even though LVEF was still within the normal range. Nevertheless, neither LVRI nor biventricular functional parameters in ART-treated males with AIDS were different from those in HCs in this study. Subgroup analysis according to different disease severities based on CD4+ cell counts and disease duration did not show significant differences. Myocardial strain reflects the change in the length of myocardial fibers in the process of contraction and relaxation, and it is an early sensitive index to evaluate subclinical cardiac dysfunction [19,20]. Case reports and cohort studies both found that ART-treated AIDS patients with a mean treatment course of 9.3 years developed significantly decreased global longitudinal strain and circumferential strain with normal LVEF [21]. Even in children and young adults with a treatment course of 6.8 years, significantly reduced global longitudinal strain was revealed [22]. The disease duration of AIDS patients in these studies was relatively longer (more than 5 years) than that of the patients in our study (within 3 years), which might be the foundation of the lack of myocardial strain observed in our study. Cardiac dysfunction is the result of a long-term cumulative effect of many factors [23,24,25,26,27,28,29,30], and the exact duration over which enough negative effects would accumulate and lead to cardiac dysfunction in AIDS needs further study.

In addition to cardiac function and myocardial strain, myocardial characterization in ART-treated males with AIDS was also performed. No evident myocardial edema or fibrosis was visually revealed by T2WI or LGE in our study. Acute HIV infection could cause myocarditis, revealed by obvious high intensity on T2WI and related symptoms, including chest pain, dyspnea, and palpitation [31]. Negative cardiac-related symptoms and myocardial injury biomarkers were consistent with the negative findings in T2WI images in our study. In the abovementioned study with a longer disease duration, LGE was identified in 24.3% of AIDS patients, and AIDS patients with positive LGE were more likely to develop adverse cardiac events (46% vs. 18%, *p* = 0.002) [18]. The relatively shorter disease course in our study might be insufficient to cause myocardial necrosis. Recent studies applying mapping sequences in AIDS found subclinical myocardial edema and fibrosis with increases in the T2 value, native T1 value, and extracellular volume [9,32]. However, detailed information about disease duration was not reported in these studies [9,32]. A mapping sequence was not performed in our study, and further research is needed to determine whether subclinical myocardial edema or fibrosis already exists in patients with a short disease duration.

Dyslipidemia is extremely common in ART-treated AIDS patients. ART is also accompanied by lipodystrophy, redistribution, and insulin resistance. All these factors would increase cardiovascular risk in AIDS patients receiving ART [33,34]. Additionally, injury caused by HIV infection, persistent inflammatory stimulation, and immune activation increase the possibility of myocardial injury [35,36]. ART is initiated immediately after the diagnosis of AIDS, and this treatment will be administered throughout life. The corresponding negative impact of ART and HIV infection-related factors on the heart will lead to significant cardiac complications. Thus, cardiovascular system monitoring is indispensable in ART-treated AIDS patients. Finally, we found that there was no significant cardiac dysfunction in ART-treated males with AIDS within 3 years, which indicated that ART was relatively safe for the heart in AIDS patients with a short disease duration. However, previous studies found obvious myocardial injury in ART-treated patients with longer disease durations. Thus, further studies focusing on ART-treated AIDS patients with disease durations longer than 3 years are also needed.

One limitation of this study was that the sample size was too small. In the next study, we will expand the sample size and perform a follow-up CMR study. Furthermore, a mapping sequence was not applied in this study. We used only the sequences that are most commonly used in the clinic for evaluation. However, the technology we used in this study is mature in clinical applications and more accessible in medical institutions.

## 5. Conclusions

ART-treated males with AIDS with a short disease duration may not develop obvious cardiac dysfunction as evaluated by routine CMR, so it is reasonable to appropriately extend the interval between cardiovascular follow-ups to more than 3 years.

## Figures and Tables

**Figure 1 diagnostics-12-02417-f001:**
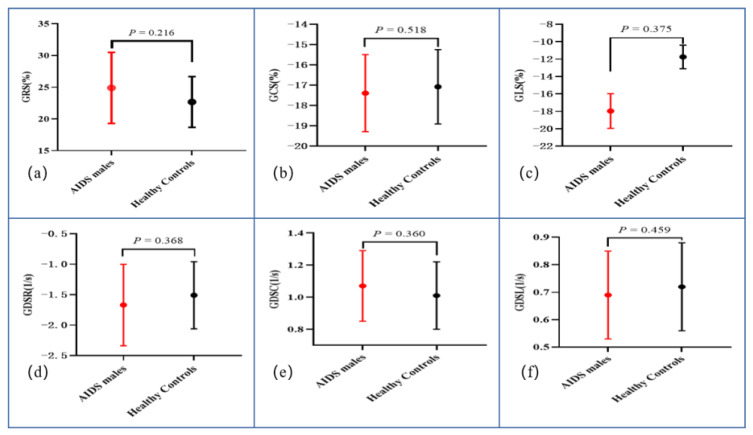
Plots for comparison of left ventricular global radial strain (**a**), global circumferential strain (**b**), global longitudinal strain (**c**), global diastolic strain rate radial(**d**), global diastolic strain rate circumferential (**e**), and global diastolic strain rate longitudinal (**f**).

**Figure 2 diagnostics-12-02417-f002:**
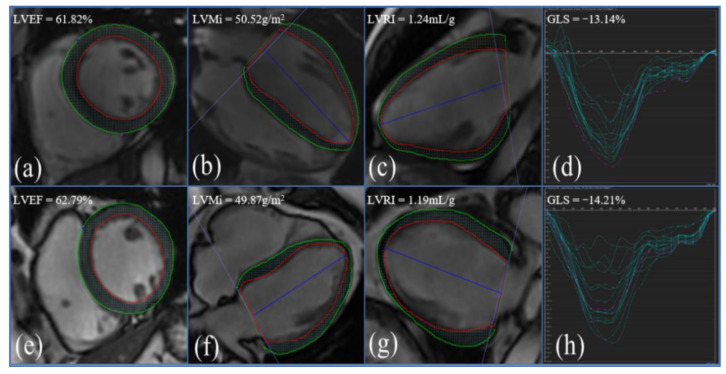
Left ventricular global longitudinal strain between ART-treated AIDS males and HCs. (**a**–**d**) AIDS, male, 27 years old. (**e**–**h**) HC, male, 30 years old. (**a**,**e**) Short axis; (**b**,**f**) four chamber; (**c**,**g**) two chamber; (**d**,**h**) GLS curve.

**Figure 3 diagnostics-12-02417-f003:**
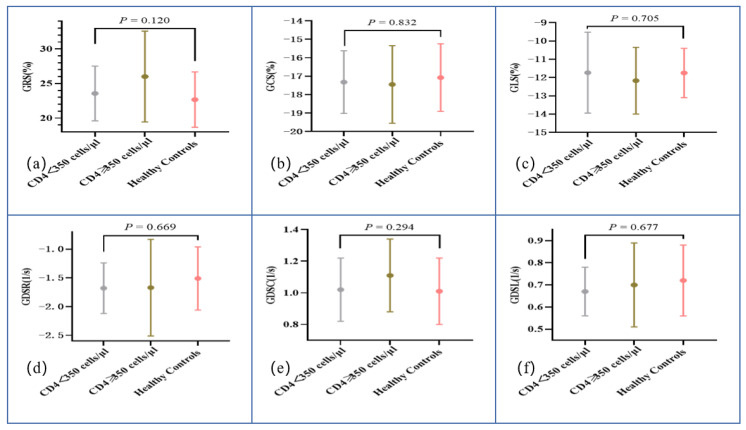
Plots for comparison of left ventricular global radial strain (**a**), global circumferential strain (**b**), global longitudinal strain (**c**), global diastolic strain rate radial (**d**), global diastolic strain rate circumferential (**e**), and global diastolic strain rate longitudinal (**f**).

**Figure 4 diagnostics-12-02417-f004:**
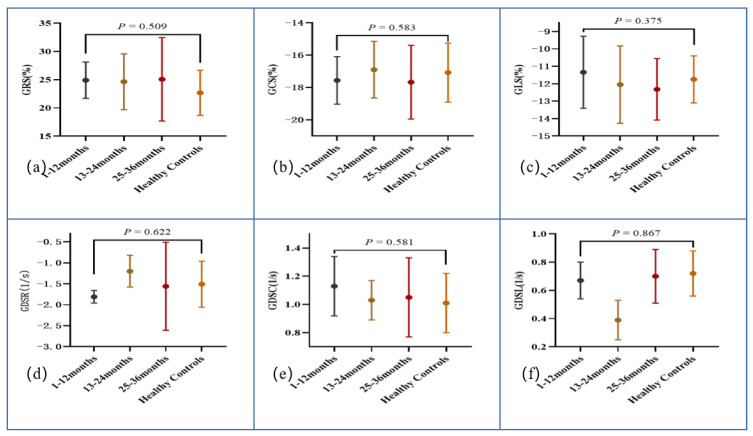
Plots for comparison of left ventricular global radial strain (**a**), global circumferential strain (**b**), global longitudinal strain (**c**), global diastolic strain rate radial (**d**), global diastolic strain rate circumferential (**e**), and global diastolic strain rate longitudinal (**f**).

**Table 1 diagnostics-12-02417-t001:** Baseline characteristics.

Characteristics	AIDS Males (*n* = 37)	Healthy Controls (*n* = 18)	*p*-Value
**General Characteristics**			
Age, y	37.62 ± 11.10	39.56 ± 11.24	0.549
Body mass index, kg/m^2^	21.88 ± 3.99	23.57 ± 2.53	0.106
Systolic BP, mmHg	126.41 ± 11.37	127.44 ± 10.03	0.743
Diastolic BP, mmHg	81.19 ± 6.79	83.11 ± 4.390	0.280
Electrocardiogram	negative		
**Risk for HIV infection**			
Heterosexual, %	19 (51.35)	MSM, %	14 (37.84)
IVDU, %	2 (5.41)	Other, %	2 (5.41)
**Complication for AIDS patients**			
Diabetes, %	0 (0.00)	Cardiovascular disease, %	0 (0.00)
Granulocytopenia, %	3 (8.11)	Metabolic syndrome, %	10 (27.03)
Erythra, %	1 (2.70)	Oral fungal infections, %	2 (5.41)
Pneumonia, %	9 (24.32)	HBV co-infection, %	3 (8.11)
HCV co-infection, %	2 (5.41)	Liver dysfunction, %	22 (59.46)
Renal dysfunction, %	1 (2.70)	Tuberculosis, %	6 (16.22)
Syphilis, %	5 (13.51)	malignant tumor, %	0 (0.00)
**Plasma metabolites in AIDS patients**			
TSHD, months	20.58 ± 3.72	CD4+ T-cell counts, cells/µL	358.21 ± 57.41
Total duration of ART, %	37 (100.00%)	CHOL, mmol/L	4.57 ± 0.68
Hs-cTnT < 3.00, ng/mL, %	37 (100.00%)	CK-MB, ng/mL	0.52 ± 0.14
Myo < 21.00, ng/mL, %	37 (100.00%)	TG, mmol/L	1.58 ± 0.11
CK-MB, μL	21.45 ± 2.79	LDH, μL	187.06 ± 19.56
HBDH, μL	159.72 ± 21.34	CK, μL	80.19 ± 5.83

Data are summarized by mean ± SD if they are normal distribution, or median (first and third quartiles) if they are abnormal distribution and *n* (%) for categorical variables. *p*-values are obtained using the Student *t*-test, or Mann–Whitney U test (for non-normal data), X^2^ test, or Fisher exact test. MSM, male who has sex with males; IVDU, intravenous drug user; TSHD, time since HIV diagnosis; ART, antiretroviral therapy; Hs-cTnT, high-sensitivity troponin T; Myo, myoglobin; CK-MB, creatine kinase isoenzyme; HBDH, hydroxybutyrate dehydrogenase; CHOL, cholesterol; TG, triglycerides; LDH, lactate dehydrogenase; CK, creatine kinase.

**Table 2 diagnostics-12-02417-t002:** CMR characteristics of ART-treated AIDS males and healthy controls.

Cardiac Function	ART-Treated AIDS Males (*n* = 37)	Healthy Controls (*n* = 18)	*p*-Value
LVEDV (mL)	122.79 ± 21.95	124.39 ± 26.18	0.706
LVESV (mL)	47.35 ± 9.30	49.73 ± 13.29	0.603
LVSV (mL)	75.43 ± 15.72	74.67 ± 15.26	0.816
LVEF (%)	61.75 ± 4.90	60.33 ± 4.66	0.307
LVM (g)	78.14 ± 15.41	85.26 ± 15.13	0.053
LVRI (mL/g)	1.59 ± 0.24	1.48 ± 0.27	0.200
RVEDV (mL)	127.78 ± 26.31	127.09 ± 24.34	0.802
RVESV (mL)	62.22 ± 18.77	62.01 ± 11.72	0.957
RVSV (mL)	65.56 ± 14.06	65.08 ± 17.38	0.513
RVEF (%)	51.94 ± 8.75	50.89 ± 6.08	0.872
LVEDVi (mL/m^2^)	68.44 ± 14.84	65.83 ± 12.55	0.720
LVESVi (mL/m^2^)	26.39 ± 6.13	26.31 ± 6.64	0.914
LVSVi (mL/m^2^)	42.05 ± 10.13	39.53 ± 7.29	0.484
LVMi (g/m^2^)	43.37 ± 8.98	45.01 ± 6.34	0.236
RVEDVi (mL/m^2^)	71.29 ± 17.19	67.41 ± 12.19	0.554
RVESVi (mL/m^2^)	34.74 ± 11.53	32.95 ± 6.29	0.788
RVSVi (mL/m^2^)	36.55 ± 8.97	34.46 ± 8.57	0.441
GRS (%)	24.89 ± 5.60	22.67 ± 4.00	0.216
GCS (%)	−17.39 ± 1.90	−17.08 ± 1.83	0.518
GLS (%)	−17.97 ± 1.99	−11.75 ± 1.35	0.375
BRS (%)	31.23 ± 6.99	30.75 ± 7.26	0.693
BCS (%)	−14.69 ± 2.20	−14.58 ± 1.78	0.673
BLS (%)	−8.73 ± 2.94	−8.86 ± 2.24	0.851
MRS (%)	24.89 ± 5.67	20.12 ± 5.08	0.360
MCS (%)	−17.30 ± 2.09	−16.44 ± 2.40	0.159
MLS (%)	−11.90 ± 2.82	−11.33 ± 2.29	0.216
ARS (%)	25.00 ± 10.02	20.94 ± 9.86	0.441
ACS (%)	−20.30 ± 2.95	−20.48 ± 2.70	0.693
ALS (%)	−14.65 ± 1.46	−14.62 ± 1.60	0.425
BDSR (1/s)	−2.43 ± 0.77	−2.05 ± 0.48	0.060
BDSC (1/s)	0.91 ± 0.19	0.92 ± 0.22	0.828
BDSL (1/s)	0.49 ± 0.43	0.62 ± 0.22	0.206
MDSR (1/s)	−1.57 ± 0.42	−1.40 ± 0.53	0.215
MDSC (1/s)	1.58 ± 0.24	1.02 ± 0.29	0.066
MDSL (1/s)	0.74 ± 0.17	0.76 ± 1.67	0.663
ADSR (1/s)	−2.06 ± 1.10	−1.84 ± 1.14	0.504
ADSC (1/s)	1.44 ± 0.40	1.37 ± 0.37	0.500
ADSL (1/s)	0.88 ± 0.17	0.85 ± 0.15	0.532
GDSR (1/s)	−1.67 ± 0.67	−1.51 ± 0.55	0.368
GDSC (1/s)	1.07 ± 0.22	1.01 ± 0.21	0.360
GDSL (1/s)	0.69 ± 0.16	0.72 ± 0.16	0.459

Data are summarized by mean ± SD if they are normal distribution or median (first and third quartiles) if they are abnormal distribution and n (%) for categorical variables. *p*-values are obtained using the Student *t*-test, or Mann–Whitney U test (for non-normal data), X^2^ test, or Fisher exact test. LVEDV, left ventricular end-diastolic volume; LVESV, left ventricular end-systolic volume; LVSV, left ventricular systolic volume; LVEF, left ventricular ejection fraction; LVM, left ventricular mass; LVRI, left ventricular remodeling index; RVEDV, right ventricular end-diastolic volume; RVESV, right ventricular end-systolic volume; RVSV, right ventricular systolic volume; RVEF, right ventricular ejection fraction; LVEDVi, left ventricular end-diastolic volume index; LVESVi, left ventricular end-systolic volume; LVMi, Left Ventricular Mass Index; RVEDVi, right ventricular end-diastolic volume index; RVESVi, right ventricular end-systolic volume; RVSVi, right ventricular systolic volume index; GRS, global radial strain; GCS, global circumferential strain; GLS, global longitudinal strain; BRS, basal radial strain; BCS, basal circumferential strain; BLS, global longitudinal strain; MRS, middle radial strain; MCS, middle circumferential strain; MLS, middle longitudinal strain; ARS, apical radial strain; ACS, apical circumferential strain; ALS, apical longitudinal strain; BDSR, basal peak diastolic strain rate radial; BDSC, basal peak diastolic strain rate circumferential; BDSL, basal peak diastolic strain rate longitudinal; MDSR, middle peak diastolic strain rate radial; MDSC, middle peak diastolic strain rate circumferential; MDSL, middle peak diastolic strain rate longitudinal; ADSR, apical peak diastolic strain rate radial; ADSC, apical peak diastolic strain rate circumferential; ADSL, apical peak diastolic strain rate longitudinal; GDSR, global peak diastolic strain rate radial; GDSC, global peak diastolic strain rate circumferential; GDSL, global peak diastolic strain rate longitudinal.

**Table 3 diagnostics-12-02417-t003:** Subgroup analysis according to the CD4+ T-cell counts.

Cardiac Function	CD4 < 350 cells/μL (*n* = 17)	CD4 ≥ 350 cells/μL (*n* = 20)	Healthy Controls (*n* = 18)	*p*-Value
LVEDV (mL)	117.04 ± 17.23	127.67 ± 24.66	124.39 ± 26.18	0.376
LVESV (mL)	46.14 ± 5.54	48.38 ± 11.65	49.73 ± 13.29	0.616
LVSV (mL)	70.89 ± 16.38	79.29 ± 14.42	74.67 ± 15.26	0.256
LVEF (%)	61.10 ± 6.18	62.30 ± 3.55	60.33 ± 4.66	0.456
LVM (g)	76.60 ± 12.60	79.44 ± 17.67	85.26 ± 15.13	0.245
LVRI (mL/g)	1.54 ± 0.19	1.64 ± 0.27	1.48 ± 0.27	0.151
RVEDV (mL)	125.18 ± 23.00	130.00 ± 29.24	127.09 ± 24.34	0.849
RVESV (mL)	61.76 ± 15.40	62.62 ± 21.63	62.01 ± 11.72	0.987
RVSV (mL)	63.42 ± 13.00	67.38 ± 14.99	65.08 ± 17.38	0.730
RVEF (%)	51.07 ± 8.63	52.67 ± 9.01	50.89 ± 6.08	0.754
LVEDVi (mL/m^2^)	64.81 ± 10.31	71.52 ± 17.49	65.83 ± 12.55	0.290
LVESVi (mL/m^2^)	25.48 ± 2.80	27.16 ± 7.96	26.31 ± 6.64	0.722
LVSVi (mL/m^2^)	39.33 ± 9.68	44.36 ± 10.16	39.53 ± 7.29	0.169
LVMi (g/m^2^)	42.25 ± 6.00	44.32 ± 10.97	45.01 ± 6.34	0.593
RVEDVi (mL/m^2^)	69.40 ± 13.86	72.90 ± 19.81	67.41 ± 12.19	0.560
RVESVi (mL/m^2^)	34.19 ± 8.69	35.21 ± 13.71	32.95 ± 6.29	0.794
RVSVi (mL/m^2^)	35.21 ± 7.98	37.69 ± 9.79	34.46 ± 8.57	0.504
GRS (%)	23.57 ± 3.96	26.01 ± 6.57	22.67 ± 4.00	0.120
GCS (%)	−17.32 ± 1.70	−17.45 ± 2.10	−17.08 ± 1.83	0.832
GLS (%)	−11.73 ± 2.21	−12.17 ± 1.82	−11.75 ± 1.35	0.705
BRS (%)	29.27 ± 6.11	32.89 ± 7.40	30.75 ± 7.26	0.292
BCS (%)	−14.74 ± 2.29	−14.64 ± 2.19	−14.58 ± 1.78	0.973
BLS (%)	−8.77 ± 2.92	−8.70 ± 3.03	−8.86 ± 2.24	0.984
MRS (%)	21.13 ± 4.55	22.53 ± 6.53	20.12 ± 5.08	0.404
MCS (%)	−16.93 ± 1.82	−17.62 ± 2.30	−16.44 ± 2.40	0.262
MLS (%)	−11.43 ± 3.45	−12.31 ± 2.17	−11.33 ± 2.29	0.467
ARS (%)	23.67 ± 6.41	26.13 ± 12.37	20.94 ± 9.86	0.288
ACS (%)	−20.54 ± 2.64	−20.09 ± 3.25	−20.48 ± 2.70	0.877
ALS (%)	−14.31 ± 1.45	−14.95 ± 1.43	−14.62 ± 1.60	0.441
BDSR (1/s)	−2.28 ± 0.60	−2.55 ± 0.88	−2.05 ± 0.48	0.088
BDSC (1/s)	0.94 ± 0.20	0.88 ± 0.19	0.92 ± 0.22	0.686
BDSL (1/s)	0.60 (0.49, 0.68)	0.58 (−0.06, 0.64)	0.59 (0.46, 0.72)	0.067
MDSR (1/s)	−1.52 ± 0.39	−1.61 ± 0.45	−1.40 ± 0.53	0.402
MDSC (1/s)	1.09 ± 0.19	1.21 ± 0.26	1.02 ± 0.29	0.063
MDSL (1/s)	0.70 ± 0.14	0.77 ± 0.19	0.76 ± 1.67	0.406
ADSR (1/s)	−1.98 ± 0.68	−2.12 ± 1.37	−1.84 ± 1.14	0.745
ADSC (1/s)	1.39 ± 0.36	1.49 ± 0.44	1.37 ± 0.37	0.581
ADSL (1/s)	0.85 ± 0.16	0.91 ± 0.18	0.85 ± 0.15	0.426
GDSR (1/s)	−1.68 ± 0.44	−1.67 ± 0.84	−1.51 ± 0.55	0.669
GDSC (1/s)	1.02 ± 0.20	1.11 ± 0.23	1.01 ± 0.21	0.294
GDSL (1/s)	0.67 ± 0.11	0.70 ± 0.19	0.72 ± 0.16	0.677

abbreviations as in Table 2. The distribution of some groups of data is not satisfied but close to the normal distribution, so the median is used to represent its concentration.

**Table 4 diagnostics-12-02417-t004:** Subgroup analysis based on disease duration.

Cardiac Function	1–12 Months (*n* = 10)	13–24 Months (*n* = 12)	25–36 Months (*n* = 15)	Healthy Controls (*n* = 18)	*p*-Value
LVEDV (mL)	116.12 ± 15.77	131.62 ± 25.00	120.16 ± 21.96	124.39 ± 26.18	0.496
LVESV (mL)	48.37 ± 5.76	48.67 ± 11.35	45.62 ± 9.72	49.73 ± 13.29	0.671
LVSV (mL)	67.75 ± 13.07	82.95 ± 16.12	74.55 ± 15.16	74.67 ± 15.26	0.176
LVEF (%)	58.05 ± 4.75	62.75 ± 3.38	63.41 ± 4.97	60.33 ± 4.66	0.050
LVM (g)	79.72 ± 14.86	79.69 ± 19.35	75.85 ± 12.84	85.26 ± 15.13	0.232
LVRI (mL/g)	1.48 ± 0.21	1.68 ± 0.23	1.60 ± 0.25	1.48 ± 0.27	0.171
RVEDV (mL)	121.65 ± 26.58	139.47 ± 28.33	122.52 ± 22.84	127.09 ± 24.34	0.398
RVESV (mL)	61.33 ± 18.59	67.36 ± 16.62	58.71 ± 20.76	62.01 ± 11.72	0.524
RVSV (mL)	60.32 ± 12.83	72.12 ± 17.40	63.81 ± 10.25	65.08 ± 17.38	0.334
RVEF (%)	50.24 ± 8.77	51.72 ± 6.78	52.23 ± 10.37	50.89 ± 6.08	0.709
LVEDVi (mL/m^2^)	65.05 ± 11.69	73.09 ± 16.82	66.97 ± 15.06	65.83 ± 12.55	0.661
LVESVi (mL/m^2^)	27.03 ± 4.11	27.01 ± 7.20	25.47 ± 6.61	26.31 ± 6.64	0.807
LVSVi (mL/m^2^)	38.02 ± 8.80	46.09 ± 10.82	41.50 ± 9.81	39.53 ± 7.29	0.254
LVMi (g/m^2^)	44.30 ± 7.46	44.15 ± 11.63	42.12 ± 7.89	45.01 ± 6.34	0.571
RVEDVi (mL/m^2^)	68.02 ± 16.36	77.56 ± 18.61	68.47 ± 16.31	67.41 ± 12.19	0.482
RVESVi (mL/m^2^)	34.13 ± 10.50	37.58 ± 10.92	32.88 ± 12.91	32.95 ± 6.29	0.500
RVSVi (mL/m^2^)	33.88 ± 8.65	39.68 ± 10.49	35.58 ± 7.49	34.46 ± 8.57	0.453
GRS (%)	24.91 ± 3.22	24.64 ± 4.94	25.07 ± 7.39	22.67 ± 4.00	0.509
GCS (%)	−17.56 ± 1.47	−16.90 ± 1.75	−17.67 ± 2.28	−17.08 ± 1.83	0.583
GLS (%)	−11.34 ± 2.07	−12.05 ± 2.22	−12.32 ± 1.77	−11.75 ± 1.35	0.600
BRS (%)	28.39 ± 6.31	32.69 ± 8.56	31.95 ± 5.84	30.75 ± 7.26	0.527
BCS (%)	−14.54 ± 1.76	−13.54 ± 2.60	−15.71 ± 1.71	−14.58 ± 1.78	0.156
BLS (%)	−8.03 ± 2.23	−8.63 ± 3.41	−9.28 ± 3.03	−8.86 ± 2.24	0.809
MRS (%)	21.94 ± 3.74	22.51 ± 1.78	21.35 ± 7.07	20.12 ± 5.08	0.598
MCS (%)	−17.39 ± 1.79	−16.97 ± 1.78	−17.51 ± 2.57	−16.44 ± 2.40	0.543
MLS (%)	−11.18 ± 3.23	−11.83 ± 2.47	−12.45 ± 2.89	−11.33 ± 2.29	0.458
ARS (%)	26.87 ± 4.26	21.69 ± 7.91	26.40 ± 13.55	20.94 ± 9.86	0.405
ACS (%)	−20.85 ± 2.67	−20.00 ± 2.57	−20.17 ± 3.57	−20.48 ± 2.70	0.744
ALS (%)	−14.26 ± 2.35	−15.02 ± 1.66	−14.62 ± 0.97	−14.62 ± 1.60	0.814
BDSR (1/s)	−2.22 ± 0.75	−2.52 ± 0.85	−2.49 ± 0.74	−2.05 ± 0.48	0.198
BDSC (1/s)	0.96 ± 0.16	0.80 ± 0.17	0.96 ± 0.20	0.92 ± 0.219	0.122
BDSL (1/s)	0.60 (0.47, 0.69)	0.52 (−0.06, 0.67)	0.60 (0.51, 0.67)	0.59 (0.46, 0.72)	0.237
MDSR (1/s)	−1.66 ± 0.34	−1.56 ± 0.30	−1.51 ± 0.55	−1.40 ± 0.53	0.549
MDSC (1/s)	1.24 ± 0.22	1.10 ± 0.16	1.15 ± 0.29	1.02 ± 0.29	0.183
MDSL (1/s)	0.76 ± 0.16	0.72 ± 0.16	0.73 ± 0.19	0.76 ± 1.67	0.914
ADSR (1/s)	−2.31 ± 0.63	−1.75 ± 0.84	−2.13 ± 1.47	−1.84 ± 1.14	0.593
ADSC (1/s)	1.52 ± 0.48	1.41 ± 0.22	1.42 ± 0.46	1.37 ± 0.37	0.812
ADSL (1/s)	0.87 ± 0.15	0.88 ± 0.13	0.89 ± 0.22	0.85 ± 0.15	0.932
GDSR (1/s)	−1.81 ± 0.15	−1.72 ± 0.38	−1.56 ± 1.05	−1.51 ± 0.55	0.622
GDSC (1/s)	1.13 ± 0.21	1.03 ± 0.14	1.05 ± 0.28	1.01 ± 0.21	0.581
GDSL (1/s)	0.67 ± 0.13	0.39 ± 0.14	0.70 ± 0.19	0.72 ± 0.16	0.867

abbreviations as in Table 2. The distribution of some groups of data is not satisfied but close to the normal distribution, so the median is used to represent its concentration.

**Table 5 diagnostics-12-02417-t005:** Correlation between CD4+ T-cell counts and CMR indices.

Cardiac Function	CD4+ T-Cell Counts (Cells/ μL)		Cardiac Function	CD4+ T-Cell Counts (Cells/ μL)	
	r	*p*-Value		r	*p*-Value
LVEDV (mL)	0.157	0.355	BCS (%)	−0.156	0.358
LVESV (mL)	0.075	0.659	BLS (%)	−0.086	0.614
LVSV (mL)	0.174	0.302	MRS (%)	−0.059	0.727
LVEF (%)	0.102	0.546	MCS (%)	0.009	0.960
LVM (g)	0.110	0.518	MLS (%)	−0.023	0.892
LVRI (mL/g)	0.025	0.883	ARS (%)	0.080	0.639
RVEDV (mL)	0.034	0.841	ACS (%)	−0.078	0.645
RVESV (mL)	0.022	0.899	ALS (%)	0.014	0.937
RVSV (mL)	0.035	0.836	BDSR (1/s)	0.014	0.936
RVEF (%)	0.009	0.958	BDSC (1/s)	−0.144	0.395
LVEDVi (mL/m^2^)	0.118	0.488	BDSL (1/s)	−0.085	0.618
LVESVi (mL/m^2^)	0.062	0.714	MDSR (1/s)	−0.069	0.684
LVSVi (mL/m^2^)	0.135	0.427	MDSC (1/s)	0.026	0.877
LVMi (g/m^2^)	0.143	0.397	MDSL (1/s)	0.013	0.941
RVEDVi (mL/m^2^)	0.027	0.875	ADSR (1/s)	0.091	0.591
RVESVi (mL/m^2^)	0.029	0.867	ADSC (1/s)	0.000	0.999
RVSVi (mL/m^2^)	0.015	0.931	ADSL (1/s)	0.041	0.810
GRS (%)	0.046	0.787	GDSR (1/s)	−0.009	0.959
GCS (%)	−0.072	0.670	GDSC (1/s)	−0.011	0.948
GLS (%)	−0.094	0.581	GDSL (1/s)	−0.088	0.603
BRS (%)	0.085	0.618			

abbreviations as in Table 2.

## Data Availability

All data generated or analyzed during the study are included in the published paper.

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
