# Peer review of "Clinical Application of Cardiac Magnetic Resonance in ART-Treated AIDS Males with Short Disease Duration"

_diagnostics, 2022, doi:10.3390/diagnostics12102417_

Round 1
Reviewer 1 Report
1. There is no uniformity in the presentation of the numerical values of a number of indicators in the tables. For example, in Table 2, the BDSC(1/s) score for the AIDS Patients group is presented as 0.91±0.19, and for the Healthy Controls group as 0.92±0.219.
2. Some indicators should be presented in tables according to their non-normal distribution. For example, in Table 3 the BDSL(1/s) in the CD4≥350 cells/μL group is presented as 0.38±0.55, and in Table 4 the same indicator for the 13-24months group is presented as 0.35±0.49.
3. For the correlation analysis presented in Table 5, it is required to use not only the Pearson test, but also the Spearman test.
Author Response
Response to Reviewer 1 Comments
Thank you very much for your comments. We answered your comments and revised the manuscript accordingly, as shown below. The revised content is marked in blue in the manuscript.
Point 1: There is no uniformity in the presentation of the numerical values of a number of indicators in the tables. For example, in Table 2, the BDSC(1/s) score for the AIDS Patients group is presented as 0.91±0.19, and for the Healthy Controls group as 0.92±0.219.
Response 1: We have checked and modified the data in the manuscript. We have unified the number of digits after the decimal point of the data. The “r” and “p” values retain three digits after the decimal point, while the others retain two digits.
Point 2: Some indicators should be presented in tables according to their non-normal distribution. For example, in Table 3 the BDSL(1/s) in the CD4≥350 cells/μL group is presented as 0.38±0.55, and in Table 4 the same indicator for the 13-24months group is presented as 0.35±0.49.
Response 2: Thank you for pointing out our problem. The distribution of BDSL in the two groups is not normal distribution. We have presented BDSL(1/S) as median and inter quartile in the two groups. And the rest data are normal distribution, so we still use mean and standard deviation to express.
Point 3: For the correlation analysis presented in Table 5, it is required to use not only the Pearson test, but also the Spearman test.
Response 3: Spearman's correlation analysis is applied to rank variables and variables with abnormal distribution. The data in TABLE 5 are continuous quantitative indicators and all of them are normal distribution, so we only used Pearson correlation analysis. When we checked the data, we found that the strain in the circumferential and long axis directions of are negative, but the negative sign represents the direction of myocardial contract. When we perform statistical analysis, we did not remove the negative sign of direction, so we used absolute value to perform statistical analysis again. The table is modified.

Reviewer 2 Report
Dear Authors,
the manuscript entitled 'Clinical application of cardiac magnetic resonance in AIDS patients receiving ART therapy with short disease duration' has the scope to evaluate cardiovascular complications in AIDS patients; CMR is an advanced imaging technique that can detect edema, LGE and with the strain also early systolic dysfunction. The study is well conducted and concluded that AIDS patients receiving ART therapy in short disease duration may not develop obvious cardiac dysfunction. Despite the small number of subjects the results has a clinical impact in continuing ART therapy in these patients to improve their quality of life.
In my opinion the manuscript can be accepted
Kind regards
Author Response
Response to Reviewer 3 Comments
Thank you very much for your comments sincerely. We answered your comments and revised the manuscript accordingly, as shown below. The revised content is marked in red in the article.
General concept comments
Point 1: The English language is not very appropriate and not always correct
Response 1: We apologize for the inconvenience caused by the poor language. We have now worked on both grammar and readability. Native English speakers are involved for language corrections.
Point 2: The discussion needs to be revised as the majority of information reported better fits the introduction section and the conclusion section.
Response 2: We agree with your opinion and have made comprehensive modifications to the introduction and discussion. In the introduction, we first briefly introduced the incidence rate and cardiovascular involvement of AIDS, and proposed that it is very important to evaluate the cardiac complications timely. Secondly, we listed the possible pathological factors leading to cardiovascular risk. Next, we introduced the advantages of CMR over echocardiography, and summarized relevant reports in CMR. However, course of disease in these studies is relative longer. Whether HIV patients receiving ART treatment have cardiac dysfunction in the early stage is unknown. Finally, based on the fact that the incidence rate of AIDS in males is much higher than that of women, and the cardiovascular risk is also higher in males, this study aims to evaluate the cardiac complications of male AIDS patients who receive ART treatment in the short term by CMR. We also made corresponding modifications in title.
In order to better fit and state our research results, we also made major changes in the discussion section. First of all, we summarized the findings of this study ---- male AIDS patients receiving ART may not have obvious cardiac dysfunction in the early stage (within 3 years). Thus the cardiac follow-up interval of AIDS patients might be appropriately extended. Secondly, we discussed the indicators of cardiac structure, function and myocardial characterization respectively. We discussed the reason of different findings between previous studies and our study. A relative shorter disease duration of our study might be insufficient to accumulate obvious myocardial injury. Besides, our study also indicate that ART is safe for the heart within 3 years. However, previous studies found obvious myocardial injury in ART treated patients with longer disease duration. Thus, further studies focus on ART treated AIDS patients with disease duration longer that 3 years are also needed.
Point 3: CD4+ T-cell counts alone cannot be used as parameters for cardiovascular damage
Response 3: We agree with you that CD4+T-cell counts is a marker of the severity of AIDS rather than a marker of myocardial injury. The main markers of myocardial injury evaluated in this study were the function and strain indicators of CMR. The patients included in this study were examined for myocardial injury marker(myocardial enzyme and troponin), and all were in the normal range. Subgroups comparison were performed according to the severity of AIDS (CD4 + T cell counts) to explore whether the severity or activity of AIDS is related to myocardial injury.
Point 4:the sample size was too small
Response 4 : We are also aware of this point and express it in the limitation section. In the early stage, these patients had no symptoms. They were mainly concerned about the severity of AIDS itself, so they did not pay enough attention to myocardial injury. Therefore, relatively fewer patients were willing to participate in the study. In the next study, we will expand the sample size and perform follow-up CMR study
Point 5: only male subjects were evaluated
Response 5 : In our country, the incidence rate of AIDS in males is far higher than that of female[1]. Besides, the cardiovascular risk of men is higher than that of women[2]. At the same time, women are always reluctant to participate in such research. Only a few female patients participated in our study. Considering that there are significant differences in cardiac function parameters between men and women, only male patients were included in order to avoid confusion[3-4].
Reference
[1] NCAIDS, NCSTD, China CDC. Update on the AIDS/STD epidemic in China in December 2017[J].Chinese Journal of AIDS & STD, 2018, 24(02): 111. DOI: 10.13419/ j. cnki. aids.2018. 02. 01.
[2] O'Neil A, Scovelle AJ, Milner AJ, Kavanagh A. Gender/Sex as a Social Determinant of Cardiovascular Risk. Circulation. 2018 Feb 20;137(8):854-864. doi: 10.1161/CIRCULATIONAHA.117.028595. PMID: 29459471.
[3]Lawton JS, Cupps BP, Knutsen AK, et al. Magnetic resonance imaging detects significant sex differences in human myocardial strain. Biomed Eng Online. 2011 Aug 22;10:76. doi: 10.1186/1475-925X-10-76. PMID: 21859466; PMCID: PMC3180436.
[4] Andre F, Steen H, Matheis P, et al. Age- and gender-related normal left ventricular deformation assessed by cardiovascular magnetic resonance feature tracking. J Cardiovasc Magn Reson. 2015 Mar 10;17(1):25. doi: 10.1186/s12968-015-0123-3. PMID: 25890093; PMCID: PMC4355347. )
Point 6:the number of healthy controls was small compared to the number of AIDS patients
Response 6 : Our health volunteers are recruited only through informed consent. Secondly, due to the epidemic situation, some people are not willing to go to hospital without any problems. We recruited healthy controls within a short term when epidemic was in well control. The two factor lead to a small sample size of healthy controls. We will increase the sample size for further research in the future.
Point 7:the cardiac magnetic resonance was performed without using the mapping sequence useful for a quantitative evaluation.
Response 7 : We agree with your opinion that mapping sequences are very useful technique and has made remarkable achievements in detecting sub-clinical fibrosis and edema. In our country, AIDS patients are distributed in cities and towns with different levels of development. Some institutions have limited access to these sequences, professional post-processing software, training corresponding personnel. Therefore, we used the most commonly accessible sequences to evaluate the cardiac complication of ART treated AIDS males. We also explained this in the limitation.
Point 8: Other limits- the introduction has to be improved, the protocol used of monitoring AIDS patients is not described: which instrumental examinations are usually adopted, at which time period they are performed? Cardiac magnetic resonance is a common diagnostic tool used in this type of patients?
Response 8 : Echocardiography is the main modality for evaluation of cardiac complication in AIDS patients for long-term follow-up with relative lower cost. However, due to its intrinsic limitation of lower accuracy of cardiac function evaluation, poor repeatability, inability of myocardial characterization, cardiac magnetic resonance (CMR) is often recommended for assessment of myocardial involvement and accurate function evaluation in AIDS patients, especially suspecting or having clinical symptoms.
Specific comments
Point 1: Pag 1,2,3,4,6,7-13 ;LGE(Figure 1): abbreviation without explanation. Figure 1 is not reported.
Response 1 : We appreciate the reviewer for pointing out this issue. We have corrected the grammar and spelling errors on each page of the article one by one according to the requirements. I'm so sorry we marked the picture incorrectly. In fact, picture 2 is the actual picture 1, and we have corrected this. We also adjusted and optimized the order of all tables and figures. The revised content is marked in red in the article.
Point 2: Impagination of pages 4, 5 and 6 has to be corrected. In all pages punctuation and spacing have to be revised. The references cited in the introduction are too remote.
Response 2 : We have revised the typesetting and punctuation in the text. At the same time, thank you for your suggestions on references. We have also introduced your proposed literature into the text.
